# Photolysis by UVA–Visible Light of TNT in Ethanolic, Aqueous-Ethanolic, and Aqueous Solutions According to Electrospray and Aerodynamic Thermal Breakup Droplet Ionization Mass Spectrometry

**DOI:** 10.3390/molecules27227992

**Published:** 2022-11-17

**Authors:** Dmitriy G. Sheven, Viktor V. Pervukhin

**Affiliations:** Nikolaev Institute of Inorganic Chemistry SB RAS, Acad. Lavrentieva Ave. 3, 630090 Novosibirsk, Russia

**Keywords:** photolysis, TNT, mass spectrometry, environmental

## Abstract

The mechanism of photolytic degradation of 2-4-6-trinitrotoluene (TNT) by UVA–visible light (>320 nm) in ethanolic, aqueous-ethanolic, and aqueous solutions was investigated by electrospray and aerodynamic thermal breakup droplet ionization mass-spectrometric analyses. For the photolysis, a DRK-120 mercury-quartz lamp was used. Products of the photolysis reaction were compared with known products of TNT transformation in the environment. Because the photochemistry of some compounds in alcohols (in contrast to aqueous solutions) features a transfer of electrons from the solvent to the light-excited compound, we believe that the efficiency of photolysis (polymerization) of TNT in ethanol and aqueous-ethanolic solutions is based on this mechanism.

## 1. Introduction

During both the production and destruction of munitions, liquid waste is produced known as “pink water” [1]. The composition of pink water depends on many factors [2], but it usually contains 2,4,6-trinitrotoluene (TNT), which has photochemical activity, giving it a pink color. TNT has toxic effects on the environment and humans [3]. Therefore, this liquid waste cannot be dumped without treatment. Waste treatment and removal and/or recycling of contaminants may involve dissolution, precipitation, biodegradation, photolysis, or other processes [4,5].

Photolysis is one of the main pathways for the transformation of TNT solutions in the environment. It is known that under the influence of sunlight, TNT enters into oxidation and reduction reactions with a release of nitrite, nitrate, and ammonium ions. From solutions in distilled water irradiated with a mercury lamp, the direct-oxidation products of TNT can be isolated, as well as nitroaniline, nitrophenol, benzisoxazole, benzaldoxime, benzonitrile, and dimerization products [6]. The rate of this photochemical degradation in water depends on the pH of the medium, and at high pH values, the degradation includes a contribution of alkaline hydrolysis [7]. To accelerate the degradation, one can implement the photocatalytic decomposition of TNT (for example, in the presence of TiO_2_) [8], use amino compounds to bind TNT in water [9], or remove TNT with perovskites, which trigger the thermal decomposition of TNT with surprisingly low activation energy barriers [10].

The influence of natural organic matter (NOM) or aquatic humic substances on the rate of TNT photolysis is interesting because such additives are inexpensive. NOM can affect the photolysis of aqueous organic pollutants by acting as a photosensitizer, for example, through the formation of reactive oxygen species (singlet oxygen and the hydroxyl radical). NOM can serve as both a source and a sink for the hydroxyl radical [11]. It has been found that humic substances raise the rate of the photodegradation of nitroaromatic compounds—as compared to the rate of the reaction in distilled water—by acting as triplet sensitizers and forming complexes (e.g., with charge transfer) [12]. On the other hand, NOM varies depending on its source, and its application may not yield consistent results. Therefore, research into the effect of solvents (as stabler compounds) on the mechanism of photolysis is important [13]. For example, the properties of the solvent (polarity and acceptor and donor properties) play a major role in the photolysis of pesticides. Experiments show a higher rate of photolysis in solvents that have lower polarity [14,15]. The solvent behaves as an electron donor, and the pesticide behaves as an electron acceptor. The degradation pathway and the end products of photolysis depend on the type of solvent in many other cases [16,17,18].

In terms of TNT photolysis, this means that the photolysis process will be more efficient in an organic solvent than in water (because the polarity of water is higher). Ethanol is a good choice of such a solvent because it is relatively inexpensive and environmentally friendly and dissolves TNT well [19]. A characteristic feature of the photochemistry of some compounds in alcohols, in contrast to aqueous solutions (for example, metal complexes [20]), is photoreduction via electron transfer from a solvent molecule to a light-excited compound. Therefore, we hypothesized a similar mechanism for TNT photolysis.

Thus, the purpose of this work was to study the kinetics and mechanism of TNT photolysis in ethanolic and aqueous-ethanolic solutions and compare the results with products of TNT phototransformation in aqueous solutions. In addition to the well-known methods of UV–vis spectroscopy and electrospray ionization (ESI) mass spectrometry, we used the recently proposed method of aerodynamic thermal breakup droplet ionization (ATBDI) for mass-spectrometric analysis [21,22], which allow one to monitor photochemical reactions in real time. Furthermore, ATBDI makes it possible to characterize the thermal degradation of photolysis products and evaluate the thermal stability of these compounds [23,24]. We hope that these data will facilitate the handling of waste containing toxic TNT-like compounds.

## 2. Materials and Methods

### 2.1. Chemicals and Reagents

Distilled water (0.2 MΩ cm) for the experiments was prepared by internal services of the Institute of Inorganic Chemistry (Novosibirsk, Russia). Ethanol of a “chemically pure” grade from Criochrom Ltd. (St. Petersburg, Russia) was used. Technical TNT was procured as part of the project “A study of the possibility of registering explosives by gas assays” carried out by the Division of Special Equipment and Communications in the Siberian Police Department (Russia). Stock solutions of TNT (10^2^ mg L^−1^) for each solvent were prepared by dissolving the substance in a solvent, followed by storage at ~4 °C. TNT was recrystallized from ethanol once before use. Solutions of required concentrations were prepared before the experiment by serial dilution. Aqueous-ethanolic solutions were prepared at a ratio of 1:1 (*v*/*v*) via the mixing of aqueous and ethanolic solutions.

TNT is a toxic substance, so personal protective equipment (gloves and goggles) should be used when working with it, and personal hygiene measures should be observed. In addition, TNT is explosive with a detonation diameter of about 1 cm (in a steel sheath); therefore, to avoid detonation, it is necessary to work with a relatively small amount of TNT (we worked with samples no more than 100 mg).

### 2.2. An ATBDI System and Thermal Degradation

The ATBDI method was implemented on an Agilent 6130 quadrupole mass spectrometer and described in detail in ref. [25] (see also Appendix A). Briefly, an analyzed sample in a polar solvent is pulverized into large neutral or singly charged droplets with a size of ~3–5 μm. The droplets obtained in the atomizer enter a heated tube (in our experiments, with an inner diameter of 1 mm, a length of 130 mm, and a temperature, T*_suction_*, varied in the range of 20–350 °C). In the tube, the droplets are heated up sharply, the surface tension of the droplets decreases, and they break down because of random ionization according to Dodd [26]. At the same time, there is intensive evaporation of the solvent and the formation of gaseous ions. As a result of this disintegration, the resultant droplets are electrically charged (without the use of high voltage or of high-energy particles), and a bipolar charged aerosol comes into being in the suction tube and enters the inlet capillary of the mass spectrometer.

In addition to ATBDI ionization in the suction tube, other processes associated with chemical reactions can occur (for instance, the thermal degradation of the analyte [23,24]). In this work, it was assumed that the ionization and chemical reactions proceeded independently of each other (i.e., a function describing the overall process is a mathematical product of functions describing each process separately). The same applies to photochemical reactions.

### 2.3. Mass-Spectrometric Conditions

All experiments were conducted on the Agilent 6130 quadrupole mass spectrometer. The following parameters were employed: nitrogen as a drying gas, temperature 350 °C, flow rate 7 L min^−1^; mass spectra were recorded in the m/z range from 100 to 1500 Da. For mass spectrometry involving ESI, a charging voltage of 4 kV was used. For ATBDI mass spectrometry, this parameter was set to zero. Often, mass spectra obtained with an ATBDI ion source contain a large number of clusters and dimeric ions. To obtain clean mass spectra, we applied the fragmentor voltage, which can be varied from 0 to 350 V. In our experiments, the fragmentor voltage was set to 100 V (unless indicated otherwise).

### 2.4. Photolysis

Photochemical reactions in solution (Appendix A) were initiated by means of an OI-18A illuminator (AO LOMO, St. Petersburg, Russia; Appendix A). The main component of the illuminator is a DRK-120 mercury-quartz lamp (Appendix A), the emission spectrum of which is shown in Appendix A. In combination with a transmission spectrum of molybdenum glass (Appendix A), the system allows one to utilize UV-A visible light (which is close to natural sunlight) for photolysis. The light used overlaps with the absorption spectrum of TNT in a water/ethanol mixture (Appendix A) in the range 300–400 nm. Note that in this setup, we did not use lenses to focus the radiation and let the light uniformly affect the entire volume of the liquid being photolyzed.

### 2.5. UV–Vis Spectroscopy Settings

Aqueous, ethanolic, and aqueous-ethanolic solutions of TNT (20 mL, 10^2^ mg L^−1^) were irradiated with illuminator OI-18A (Appendix A) for 30 min. During this irradiation, 0.5 mL samples of each solution were collected after 0, 9, 18, and 30 min of the irradiation. Absorption spectra were determined by means of a quartz cuvette with 10 mm light path and an Agilent 1260 Infinity Diode Array and Multiple-Wavelength Detector.

## 3. Results and Discussion

### 3.1. Changes in Appearance and UV–Vis Spectra during the Photolysis of TNT in Aqueous and Ethanolic Solutions

The effect of the solvent on the mechanism of photochemical degradation of TNT can be predicted from the appearance of photolyzed solutions of TNT in various solvents. Figure 1 (top inset) shows a photograph of TNT solutions (10^2^ mg L^−1^) subjected to UVA–visible photolysis for 30 min (a: the aqueous solution; b: the aqueous-ethanolic solution, 1:1, *v*/*v*; and c: the solution in ethanol). During the photolysis, if we compare the solvents tested, the aqueous-ethanolic solution of TNT acquired a red-pink color faster than the other two solutions did under the same photolysis conditions. On the other hand, upon photolysis, the solution of TNT in pure ethanol stained weakly (acquired a yellowish color). Accordingly, the UV–vis absorption spectra of the TNT solutions subjected to photolysis changed, which is evident in Figure 1A–C (in these spectra, the absorption of light is indicated as a percentage of the maximum absorption of TNT in a mixture of water and ethanol at a wavelength of 227 nm, see Appendix A). Alterations of the UV–vis spectrum during photolysis occurred at wavelengths 350–600 nm (Figure 1). In the aqueous solution, a peak appeared in the region of ~505 nm (Figure 1A), which constantly grew during the photolysis (up to an hour in our experiments). This wavelength is characteristic of Meisenheimer complexes of TNT with various amines [27,28]. The peak at ~505 nm was also attributed to a Meisenheimer complex in our recent report [29]. In contrast, the photolysis of the TNT solution in ethanol did not result in a 505 nm peak in the UV–vis spectra (Figure 1C). Instead, we observed a broadening of the main peak presented in Appendix A. Apparently, the emergence of photolysis products that are structurally similar to the TNT molecule (and therefore absorb light in the same wavelength region) is responsible for the observed broadening. In the aqueous-ethanolic solution (water/ethanol, 1:1, *v*/*v*), an intermediate pattern was registered (Figure 1B). The broadening of the main peak was still observed, although it was less pronounced than that in pure ethanol. The peak at 505 nm was barely visible: we had to increase the photolysis time from 18 to 30 min to distinguish this peak on the slope of the main peak of the UV–vis spectrum of TNT (Figure 1B, curve 4). Note that an increase in the photolysis time of the TNT solution in pure ethanol did not have such an effect (Figure 1C, curve 4).

### 3.2. ESI Mass Spectra of TNT Solutions Subjected to Photolysis in Water and Ethanol

The appearance of the solutions and their UV–vis spectra correlated with the ESI mass spectra of these solutions. Figure 2 depicts the mass spectra of the TNT solutions (10^2^ mg L^−1^) subjected to UVA–visible photolysis (a: the aqueous solution; b: the aqueous-ethanolic solution, 1:1, *v*/*v*; and c: the solution in ethanol). The photolysis was carried out for 20 min. The photolysis of TNT in ethanol and in the water/ethanol mixture gave a richer mass spectrum than did the photolysis of the aqueous solution of TNT, indicating more effective photodegradation of TNT in the presence of ethanol. In agreement with our previous article [27], the aqueous solution of TNT showed only two peaks during the photolysis: at m/z 226, (TNT—H)^−^ and at m/z 453 (2TNT—H)^−^, which intensified with photolysis time. Taylor et al. [30] have proposed that the photolysis of TNT produces the complex depicted in Figure 1 (mol. wt. 454); this complex then loses water. We assigned the peak at m/z 453 to the formation of a Meisenheimer complex with an amine [27]. It should be noted that under the conditions of our experiments, the TNT dimer weakly underwent further decomposition in the aqueous solution (Figure 2A), whereas the TNT dimer was not detectable in ethanol, but a complex of two TNT molecules without H_2_O (m/z = 435) was clearly present (Figure 2C), in full accordance with Figure 1.

The photoreduction of the TNT dimer can be explained by the electron transfer from the ethanol molecule to the light-excited TNT dimer; this process is characteristic of alcoholic solutions of metal complexes [20]. It can be expected that the aqueous-ethanolic solution donates electrons less readily; therefore, in this solution, instead of the peak with *m*/*z* 435, we see a peak with *m*/*z* 437 (Figure 2B), which corresponds to the loss of an oxygen atom instead of the water molecule.

Figure 2B and c shows many more peaks, some of which can be explained with the help of data from the literature. For example, the peak with *m*/*z* 390 corresponds to several dimeric TNT compounds (with a molecular weight of 391) arising via reduction and transforming into one another under the action of light (Figure 2) [7].

Similarly, the peak at *m*/*z* 406 can be assigned to azoxy and azo TNT dimers [7] (Figure 3a). As for the peak with *m*/*z* 419, no suitable TNT photodegradation product could be found in the literature. Nonetheless, 2,2′-dicarboxy-3,3′,5,5′-tetranitroazoxybenzene is present among the photolysis products [31] (Figure 3b, molecular weight 466). Denitration of this compound during the photolysis in ethanol may have yielded a peak with *m*/*z* 419 in the mass spectrum (Figure 3b).

Figure 2 shows peaks that we failed to identify (e.g., *m*/*z* 599, 628, and 911). These peaks are suggestive of further TNT polymerization during the photolysis. Polymerization of TNT is a well-known process in the photolysis of TNT in the environment (e.g., in natural bodies of water) [31]. According to the intensity of the peaks in Figure 2A–C, the presence of ethanol accelerated the polymerization process.

The ESI mass spectra of the TNT solutions subjected to UVA–visible photolysis changed with photolysis time. In Appendix A, the kinetics of such a change are presented using the solution of TNT in ethanol as an example. On the other hand, ESI does not allow one to monitor alterations in a mass spectrum in real time: it is necessary to take a sample after photolysis, prepare it for analysis, and introduce it into the mass spectrometer; this whole procedure takes tens of minutes. Therefore, we chose the ATBDI system to study the UVA–visible photolysis of TNT (in the ethanol solution) and the thermal degradation of photolysis products.

### 3.3. Mass Spectra of ATBDI after the Photolysis of the TNT Solutions in Ethanol

Figure 3 shows changes of the ATBDI mass spectra for TNT in ethanol upon photolysis by UVA–visible light depending on photolysis time. It is obvious how the deprotonated TNT peak (*m*/*z* = 226) and its ethanol clusters [(TNT—H)^−^ and (TNT—H)^−^(C_2_H_5_OH)_n_] disappear from the mass spectrum within several minutes of photolysis. At the same time, a peak with m/z 390 and its ethanol clusters emerge in the spectrum, technically corresponding to a TNT dimer that lost NO_2_ and H_2_O. Structures of several dimeric TNT compounds that can have a peak at *m*/*z* 390 are displayed in Figure 2. The photolysis of the sample did not end there. At 14.5 min of photolysis (Figure 3c), a deprotonated-TNT peak (*m*/*z* = 226) disappeared completely, and a peak with *m*/*z* 570 acquires noticeable intensity in the spectrum. This peak technically corresponds to a TNT trimer that lost two NO_2_ groups, H_2_O, and a proton. This means that in ethanol, under the action of UVA–visible light, TNT polymerized rather rapidly, and under the influence of light, the peak with *m*/*z* 570 also gradually disappeared from the spectrum.

Figure 4 shows the time dependence of ATBDI mass spectrometric peaks for the solution of TNT in ethanol (10^2^ mg L^−1^) during the photolysis by UVA–visible light. The intensities of the TNT monomer, dimer, and trimer successively diminished with photolysis time, and these compounds transformed into ever longer polymer chains. This finding contradicts a study on a solution of TNT in water during UVA–visible photolysis [30], where a TNT dimer (apparently a Meisenheimer complex) remained stable for tens of minutes. The polymerization of TNT in the presence of light in our ethanol solution can be explained by the transfer of an electron from an ethanol molecule to a light-excited TNT molecule, followed by its conglomeration with other TNT molecules.

The ATBDI mass spectra displayed in Figure 3 differs from the ESI mass spectra (Figure 2C). The ESI spectrum is richer and contains peaks that are absent in the ATBDI mass spectra: *m*/*z* = 406, 419, and 435. In fact, these peaks can also be obtained in the ATBDI mass spectra by varying the temperature of the suction tube (T*_suction_*, Appendix A). These peaks are clearly visible in the ATBDI mass spectra presented in Appendix A for TNT in ethanol at different T*_suction_* values. The sensitivity of the mass spectra of ATBDI TNT in ethanol to the suction tube temperature points to instability of the solution under the influence of this external factor. Therefore, the time required to collect and prepare a sample for ESI mass spectrometry after the photolysis can distort the results. This disadvantage is virtually absent when ATBDI mass spectrometry is used, where we set the suction tube temperature to 242 °C because these conditions gave the simplest easy-to-interpret mass spectrum.

To complete the picture, we conducted experiments on the photolysis of TNT in ethanol (10^2^ mg L^−1^) at shorter wavelengths (λ > 200 nm). To this end, we replaced the molybdenum glass photolyzable liquid reservoir (Appendix A) with a quartz flask. The threshold of light transmission by quartz is in the shorter wavelength region (>200 nm, Appendix A). Thus, the sample was exposed to a quartz lamp with λ < 300 nm (Appendix A). The results on ATBDI mass spectra alterations during the photolysis are shown in Appendix A. Obviously, the photolysis outcomes were not influenced qualitatively by the wavelength (compare Appendix A and Figure 3); however, peaks corresponding to the TNT trimer were relatively more intense during exposure to the light of shorter wavelengths (Appendix A), and there was a peak presumably containing four TNT molecules (that lost NO_2_ and water; Appendix A) in the mass spectrum. Consequently, in our experiments, the decrease in the irradiation length accelerated the polymerization of the TNT dissolved in ethanol.

## 4. Conclusions

It was experimentally demonstrated that the photolysis of TNT in solution depends on the nature of the solvent (water, water/ethanol mixture, or ethanol). The outcomes of the photolysis differ both visually and in the UV–vis spectra and mass spectra (both in the standard ESI mass-spectrometric analysis and in the recently proposed ATBDI mass-spectrometric analysis). Overall, ethanol in a TNT solution promotes TNT polymerization during near-ultraviolet light (UVA–vis) photolysis, and polymerization efficiency increases with the decreasing wavelength of the irradiating light. Because the photochemistry of some compounds in alcohols (in contrast to aqueous solutions) features a transfer of electrons from the solvent to the light-excited compound, we can theorize that the efficiency of the photolysis (polymerization) of the TNT in ethanolic and aqueous-ethanolic solutions is based on this mechanism.

Because photolysis is one of the main pathways for the transformation of TNT solutions in the environment, this knowledge can be applied to the treatment of waste containing toxic TNT-like compounds.

## Data Availability

The raw data will be available from the corresponding author upon reasonable request.

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
