# Peer review of "Photolysis by UVA–Visible Light of TNT in Ethanolic, Aqueous-Ethanolic, and Aqueous Solutions According to Electrospray and Aerodynamic Thermal Breakup Droplet Ionization Mass Spectrometry"

_molecules, 2022, doi:10.3390/molecules27227992_

Round 1

Reviewer 1 Report

The manuscript "Photolysis by UVA-visible light of TNT in ethanolic, aqueous-ethanolic, and aqueous solutions according to electrospray and aerodynamic thermal breakup droplet ionization mass spectrometry" is devoted to the study of trinitrotoluene photolysis using two techniques: UV-Vis spectroscopy and mass spectrometry. Authors had shown that the course of the chemical processes depend on the solvent used and identified the plausible products of photodegradation of TNT. The paper would be of help for those who work in the field of disposal of explosives and criminalists.

There are some minor comments:

1. TNT is a toxic compound and explosive. Please, add some information to the Experimental section about precaution measures that should be considered during the work with TNT.

2. Lines 86-88 are evidently abundant and should be deleted.

3. Despite the high quality of paper (I especially liked the ESI file providing the very detailed description of the experimental setup), it leaves a feeling of understatement upon reading. There are kinetic curves (dependencies intensities of different peaks in mass spectra on time) that could easily be recalculated to the current concentrations; and rate constants could be deduced. Why not to do that?

4. Why the spectra on Fig. 1B, C are noisy?

Author Response

  1. TNT is a toxic compound and explosive. Please, add some information to the Experimental section about precaution measures that should be considered during the work with TNT.

Response

We added Safety precautions to the Experimental - TNT is a toxic substance, so personal protective equipment (gloves and goggles) should be used when working with it, as well as personal hygiene measures should be observed.
In addition, TNT is explosive with a detonation diameter of about 1 cm (in a steel sheath), therefore, to avoid detonation, it is necessary to work with a relatively small amount of TNT (we worked with samples no more than 100 mg).

  1. Lines 86-88 are evidently abundant and should be deleted.

Response

Accepted

  1. Despite the high quality of paper (I especially liked the ESI file providing the very detailed description of the experimental setup), it leaves a feeling of understatement upon reading. There are kinetic curves (dependencies intensities of different peaks in mass spectra on time) that could easily be recalculated to the current concentrations; and rate constants could be deduced. Why not to do that?

Response

Indeed, in some cases, our setup can be used to determine the kinetic parameters of the reaction (see references [23, 24]). Unfortunately, the determination of kinetic parameters in the case of photochemical reactions requires a separate study. In particular, it is necessary to determine the number of photons absorbed by the target compound. This parameter will depend on both the irradiation wavelength and the irradiation time (since the products of photochemical reactions will interfere with the interaction of light with the target compound).

  1. Why the spectra on Fig. 1B, C are noisy?

Response

We think that the spectra in Fig. 1B, C are noisy because they correspond to several compounds (with m/z 390, 406, 419, etc. see Fig. 2B, C). In contrast to this, Fig. 1A is relatively quiet and corresponds to one compound (m/z 453 see Fig. 2 A).

Reviewer 2 Report

Comments on molecules-2037396, “Photolysis by UVA-visible light of TNT in ethanolic, aqueous-ethanolic, and aqueous solutions according to electrospray and aerodynamic thermal breakup droplet ionization mass

spectrometry”, by Sheven & Pervukhin

This manuscript describes an experimental study on photolysis of TNT using UVA light. Products are identified using mass spectrometry and also kinetics. Extensive polymerization was observed upon photo-reduction. Mechanism is also discussed based on the product observation. The manuscript was well written. Publication is recommended.

 Comments:

 # Is there other reason to study the photolysis in organic solutions used here? In environment, water is almost the only solvent. Is this type of study for waste treatment before discharging to the environment?

 # I noticed that the photolysis was done with high concentration of TNT up to 100 mg/L. There might be a chance to separate the intermediate products using some sort of chromatograph technique and to definitely determine the structures of the products. This might help.

 # Is there concentration-dependent data on the kinetics? Ratios of polymerization depend on TNT concentration. Would polymerization be possible under some “ambient” environmental conditions?

Author Response

 Comments:

 # Is there other reason to study the photolysis in organic solutions used here? In environment, water is almost the only solvent. Is this type of study for waste treatment before discharging to the environment?

Response

In our opinion, the treatment of TNT waste prior to release to the environment is the most interesting for applying the results of this study in practice (because remediation with ethanol is very expensive). Note, however, that environmental water always contains impurities (in particular, humic compounds) that can contribute to the decomposition of TNT. The article can contribute to the search for such compounds.

 # I noticed that the photolysis was done with high concentration of TNT up to 100 mg/L. There might be a chance to separate the intermediate products using some sort of chromatograph technique and to definitely determine the structures of the products. This might help.

Response

In fact, the structures of the products were determined by GC/MS, HPLC/MS, and NMR methods in a number of works [30–33]. We used these literature data to interpret our results.

 # Is there concentration-dependent data on the kinetics? Ratios of polymerization depend on TNT concentration. Would polymerization be possible under some “ambient” environmental conditions?

Response

If dimerization (polymerization) according to Scheme 1 is the main way of the photochemical reaction, the m/z 226 curve in Fig. 4 must obey the reaction order above the first. This will cause the polymerization coefficients to be dependent on the concentration of TNT. Unfortunately, we were unable to determine the reaction order from the data in Fig. 4. A possible reason is the formation of photochemical reaction products that interfere with the decomposition of TNT.

With regard to polymerization under ambient conditions, such polymerization is possible when exposed to rainwater and sunlight [31] as well as when exposed to sea water and sunlight [33] on TNT. Of course, under these conditions, polymerization is not the only way of TNT degradation.

Reviewer 3 Report

The authors studied the Photolysis by UVA-visible light of TNT in ethanolic, aqueous-ethanolic, and aqueous solutions. The work is interesting to readers in the field of environmental chemistry and analytical chemist. Overall, the manuscript is interesting, and the experiments seem to be carried out carefully. I accept this manuscript in the current form. 

Author Response

with best wishes